# Gamma oscillations point to the role of primary visual cortex in atypical motion processing in autism

Elena V. Orekhova[1]*, Viktoriya O. Manyukhina[1,2], Ilia A. Galuta[1], Andrey O. Prokofyev[1], Dzerassa E. Goiaeva[1], Tatiana S. Obukhova[1], Kirill A. Fadeev[1], Justin F. Schneiderman[3], Tatiana A. Stroganova[1]

**1** Center for Neurocognitive Research (MEG Center), Moscow State University of Psychology and Education, Moscow, Russian Federation, **2** National Research University Higher School of Economics, Moscow, Russian Federation, **3** MedTech West and the Institute of Neuroscience and Physiology, Sahlgrenska Academy, The University of Gothenburg, Gothenburg, Sweden

* Orekhova.elena.v@gmail.com

**Data Availability Statement:** All relevant data are within the manuscript and its Supporting Information files.

## Abstract

Neurophysiological studies suggest that abnormal neural inhibition may explain a range of sensory processing differences in autism spectrum disorders (ASD). In particular, the impaired ability of people with ASD to visually discriminate the motion direction of small-size objects and their reduced perceptual suppression of background-like visual motion may stem from deficient surround inhibition within the primary visual cortex (V1) and/or its atypical top-down modulation by higher-tier cortical areas. In this study, we estimate the contribution of abnormal surround inhibition to the motion-processing deficit in ASD. For this purpose, we used a putative correlate of surround inhibition–suppression of the magnetoencephalographic (MEG) gamma response (GR) caused by an increase in the drift rate of a large annular high-contrast grating. The motion direction discrimination thresholds for the gratings of different angular sizes (1˚ and 12˚) were assessed in a separate psychophysical paradigm. The MEG data were collected in 42 boys with ASD and 37 typically developing (TD) boys aged 7–15 years. Psychophysical data were available in 33 and 34 of these participants, respectively. The results showed that the GR suppression in V1 was reduced in boys with ASD, while their ability to detect the direction of motion was compromised only in the case of small stimuli. In TD boys, the GR suppression directly correlated with perceptual suppression caused by increasing stimulus size, thus suggesting the role of the top-down modulations of V1 in surround inhibition. In ASD, weaker GR suppression was associated with the poor directional sensitivity to small stimuli, but not with perceptual suppression. These results strongly suggest that a local inhibitory deficit in V1 plays an important role in the reduction of directional sensitivity in ASD and that this perceptual deficit cannot be explained exclusively by atypical top-down modulation of V1 by higher-tier cortical areas.

**Funding:** This study was supported by Russian Science Foundation (project # 22-25-00419, to EVO). The funder had no role in study design, data collection and analysis, decision to publish, or preparation of the manuscript.

**Competing interests:** The authors have declared that no competing interests exist.

## 1. Introduction

Autism spectrum disorders (ASD) are characterized by impairments in social interaction, communication, and restricted patterns of behavior, interests, or activities. Most people with ASD also experience sensory problems [1–5]. Genetic, neurochemical, and electrophysiological studies strongly implicate an imbalance between excitation and inhibition as a common basis for sensory abnormalities in ASD [6–12].

A remarkable feature of people with autism is atypical processing of visual motion [13–15]. However, motion direction discrimination paradigms used in ASD individuals have revealed incongruent findings, i.e., impaired sensitivity to small high-contrast patterns [16, 17] vs. superior ability to discriminate the direction of motion–especially for large stimuli [18, 19]. These discrepant findings may indicate substantial interindividual variation in the genetically and neurobiologically heterogeneous ASD population [20].

From the neurophysiological perspective, at least some features of atypical motion perception in ASD could originate from disturbed center-surround antagonism in the primary visual cortex (V1) [16, 17]. Indeed, neurophysiological studies highlight the pivotal role of surround inhibition in V1 in discriminating the direction of motion of small-size objects [21, 22] while top-down modulation of V1 by higher-tier visual areas is implicated in perceptual suppression of background-like motion of large objects [23].

The role of atypical inhibition in V1 in visual perception in ASD is supported by magnetic resonance spectroscopy (MRS) [24]. However, two recent studies, which used fMRI and computer modeling, suggest a high-level origin of atypical motion perception in ASD and normal neuronal representations of moving stimuli in V1, at least in adults with average/above-average IQ [19, 25]. More direct electrophysiological measures of neural excitation-inhibition interactions in the V1 could help resolve this controversial issue.

Oscillatory gamma response (GR) elicited by visual stimulation is a potentially useful signature of excitation-inhibition balance (E-I balance) in the human brain [26]. Cortical gamma oscillations are generated by local excitatory-inhibitory loops and their synchronization crucially depends on the E-I ratio in cortical networks [27]. The strong dependence of visual GR on the parameters of sensory stimulation [26, 28–31] may explain the discrepant results regarding gamma-range MEG/EEG abnormalities found in ASD [32]. On the other hand, this dependence on stimulation may be informative with respect to putative E-I imbalance in ASD.

Animal studies have shown that increasing excitatory drive by increasing the intensity of stimulation increases both E and I, but gradually changes the E-I ratio in the V1 towards inhibition [33, 34]. Such a decrease in E-I ratio under strong stimulation may lead to suppression of induced gamma oscillations, even if neuronal firing increases or does not change [31]. In a series of MEG studies in humans, we manipulated the drift rate of a visual grating. Increasing gratings' drift rate from 0 to ~8–10 Hz leads to an increase of neuronal firing [31] and gamma oscillation frequency [31, 35, 36], which collectively points to a growing net excitation. Therefore, and similar to luminance contrast [29, 37], drift rate can be seen as a proxy for excitatory drive. This raises the question of why growing velocity/drift rate first increases GR amplitude and then, after a certain transition point, leads to GR suppression [35]. 'Tuning' of V1 neurons to certain visual motion velocities cannot explain the additive effect of contrast and velocity on GR magnitude, as we discussed elsewhere [38]. On the other hand, this additive effect agrees well with modeling studies that suggest a suppressive effect of 'excessive' excitatory drive on induced GR due to a disproportionate increase in inhibitory transmission [39]. The suppressive effect of growing inhibition explains our previous results on the association between the magnitude of GR suppression with increasing grating drift rate and individuals' performance

in a visual perception task [40] and their visual sensitivity [41, 42]. We, therefore, suggested that GR suppression might serve as an index of inhibition strength in the visual cortex.

Herein, we attempt to reveal the role of putative neural inhibitory deficits in atypical motion perception in children with ASD by exploring velocity-related modulation of the visual GR. We used two separate sets of high-contrast drifting gratings optimized for the respective purposes and presented them in two separate experiments (psychophysical and MEG). To assess motion direction discrimination thresholds and the perceptual effect of spatial suppression, we used a well-established experimental design that has been previously applied for this purpose in healthy subjects and individuals with neuropsychiatric disorders [16–18, 43–45]. To this end, we presented participants with large and small vertical gratings that drifted left- or rightward for a short period of time. The main reason for not using the same stimulus design in the neurophysiological experiment was that very brief stimulus presentation, which is crucial for estimating motion duration thresholds, does not produce GR in visual cortex in adults [46]. In addition, the magnitude of GR is strongly reduced when small (1° of the visual angle) stimuli are used because they do not engage the surround inhibition [47]. The use of 'optimal' stimuli to detect reliable GR is especially important in children, who have lower gamma oscillations power than adolescents and adults [35, 48]. Therefore, to induce GR in MEG, we used large high-contrast annular inward drifting gratings with long presentation times. By changing the drift rate of these gratings ('slow', 'medium', 'fast'), we were furthermore able to assess the velocity-related suppression of the GR as a putative index of inhibition strength [38, 40–42]. MEG data were obtained from 42 children with ASD and 37 typically developing (TD) children. Psychophysical data were available in 33 and 34 of these participants, respectively.

We expected that the shape of the GR modulation curve might provide a neural marker for weakened inhibitory signaling in V1 networks in ASD. Given that the local inhibition in V1 has a causative effect on the tuning of receptive fields [49, 50] and regulates directional sensitivity of nearby neurons [50], we also predicted that an attenuated GR suppression in V1 would be specifically linked to poor motion direction sensitivity to small-size stimuli in children with ASD.

## 2. Materials and methods

### 2.1. Participants

Initially, 45 TD boys and 63 boys with ASD aged 7–15 years were enrolled into the study. We limited our sample to males because the ratio of males to females among people with ASD is very high (~4/1) [51], and the relatively small sample size of our study would not have allowed us to analyze female participants with ASD as a separate group. On the other hand, the putative differences in neural excitability between males and females with ASD [52] preclude combining them into a single sample. The TD children were recruited from local schools. The participants with ASD were recruited from rehabilitation centers affiliated with the Moscow University of Psychology and Education. The ASD diagnosis was confirmed by an experienced psychiatrist and was based on the Diagnostic and Statistical Manual of Mental Disorders (5th ed.) [53] criteria as well as an interview with the parents/caregivers. In addition, parents/caregivers of all children were asked to complete the Russian translation of the Social Responsiveness Scale (SRS) for children [54]. None of the TD participants had known neurological or psychiatric disorders. All participants had normal or corrected to normal vision. IQ has been evaluated through standard scores on K-ABC subscales (Simultaneous and Sequential), as well as by calculating the Mental Processing Index (MPI) [55]. Before administering each K-ABC sub-test, testers always made sure that the child was following verbal instructions.

We analyzed properties of visual gamma oscillations in those children who had sufficient amount of artifact-free MEG data and structural MRI of the brain. Seven TD children were

**Table 1. Characteristics of the pediatric samples included in analysis of MEG visual gamma oscillations.**

|  | ASD | TD | T | p |
|---|---|---|---|---|
|  | mean±SD (range) | mean±SD (range) |  |  |
| Age | 10.3±2.1 | 11.0±2.0 | 1.67 | 0.11 |
| $N_{TD/ASD}$ = 37/42 | (7.5–15.3) | (7.3–15.5) |  |  |
| Sequential IQ | 80.4±17.5 | 106.1±11.1 | 7.65 | <1e-6 |
| $N_{TD/ASD}$ = 37/42 | (49–127) | (88–131) |  |  |
| Simultaneous IQ | 89.2±20.6 | 120.9±13.8 | 7.91 | <1e-6 |
| $N_{TD/ASD}$ = 37/42 | (58–144) | (91–150) |  |  |
| MPI Standard IQ | 84.6±19.3 | 119.4±11.6 | 9.56 | <1e-6 |
| $N_{TD/ASD}$ = 37/42 | (54–127) | (94–141) |  |  |
| SRS | 108.1±20.7 | 46.7±24.1 | -11.70 | <1e-6 |
| $N_{TD/ASD}$ = 35/38 | (57–145) | (8–104) |  |  |

'$N_{TD/ASD}$' indicates the number of participants in the corresponding groups for whom data were available. T and p denote Student's t-statistics and the probability of $H_0$, respectively.

excluded from the analysis because of excessive muscle or motion artifacts during the MEG acquisition, which resulted in fewer than 40 artifact free epochs per condition or, because of a lack of a technically sound MRI scan, which did not allow the generation of a proper individual head/brain model for source localization. One TD child was excluded because he did not press a button during the MEG experiment. The resulting TD sample included 37 children. Eighteen children from the ASD sample were excluded because of excessive muscle or movement artifacts during MEG recording or because it was impossible to obtain an artifact-free MRI of the brain. In addition, one child was excluded because of a technical problem during the MEG recording and two more were excluded because they failed to press a button during MEG experiment. The resulting ASD sample comprised 42 children. The 21 excluded children with ASD were significantly younger (mean age 9.17 years vs. 10.3 years; $T_{(61)}$ = 2.20, p = 0.032), and tended to have lower MPI IQ (included: 87.8, excluded: 81.7; $T_{(56)}$ = 1.26, p = 0.21; the MPI scores were available in 16 of 21 excluded ASD children). Table 1 shows the characteristics of the TD and ASD children included in the analysis of gamma oscillation parameters. The majority of the subjects in the present study participated in our previous studies in which visual gamma oscillations were analyzed in sensor space [40, 56] and/or in the behavioral study of visual motion perception [17].

Correlations between gamma parameters and performance in the psychophysical experiments were analyzed in the subset of children for whom results of those tests were available and who also had reliable visual GRs in at least one of the experimental conditions (31 of 37 TD and 26 of 42 ASD children; see below for the description of the experimental conditions and GR reliability criteria). Table 2 presents the characteristics of the children included in the analysis of correlations between MEG and psychophysical parameters.

ASD children scored significantly higher on the SRS and had significantly lower IQ than the TD children (Tables 1 and 2). Note that although we report sequential and simultaneous IQs for descriptive purposes, only MPI IQ was used for statistical analyses hereafter. Variability in MPI scores was high in the ASD sample and ranged from 'very low' to 'higher than average'. Given that about 50% of people with ASD have an IQ below 85 [57], the sample can be considered representative of the ASD spectrum.

The Ethical Committee of the Moscow University of Psychology and Education approved this investigation. All children provided their verbal assent to participate in the study and their caregivers provided written consent to participate.

**Table 2. Characteristics of the pediatric samples included in the analysis of correlations between MEG and psychophysical parameters.**

| | ASD | TD | T | p |
|---|---|---|---|---|
| | mean±SD (range) | mean±SD (range) | | |
| Age | 10.5±2.2 | 11.3±2.0 | 1.49 | 0.14 |
| $N_{TD/ASD}$ = 31/26 | (7.5–15.3) | (7.7–15.5) | | |
| Sequential IQ | 84.1±16.2 | 106.0±11.6 | 5.95 | <1e-6 |
| $N_{TD/ASD}$ = 31/26 | (49–112) | (88–131) | | |
| Simultaneous IQ | 90.6±20.5 | 121.3±14.9 | 6.53 | <1e-6 |
| $N_{TD/ASD}$ = 31/26 | (58–144) | (91–150) | | |
| MPI Standard IQ | 87.4±19.2 | 118.2±11.5 | 7.45 | <1e-6 |
| $N_{TD/ASD}$ = 31/26 | (56–127) | (94–138) | | |
| SRS | 110.3±19.0 | 46.3±24.8 | -10.04 | <1e-6 |
| $N_{TD/ASD}$ = 29/22 | (76–141) | (8–104) | | |

In all but one participant (TD), the MEG and psychophysical experiments were conducted on different days. In 23 of 37 TD children and in 26 of 42 children with ASD, the MEG experiment was performed first.

## 2.2. Psychophysical experiment

A graphical overview of the psychophysical experiment is presented in Fig 1A. Visual stimuli were presented using PsychToolbox [58]. The stimuli consisted of a vertical full-contrast sine-wave grating (1 cycle/° of visual angle) that drifted at a constant speed of 4°/s. The size of the stimuli was controlled by a two-dimensional Gaussian envelope whose full-width at half-maximum was set to 1 or 12° of visual angle for the small and large stimuli types, respectively. The direction of visual motion (left or right) was determined randomly for each trial. Participants' eyes were ~60 cm from the monitor on which stimuli were presented (Benq XL2420T, 24" LED, 1920 x 1080 resolution, 120 Hz).

An automated staircase procedure was used to estimate the minimum exposure time required for the subject to distinguish the direction of visual motion. Participants were asked to make an un-speeded two-alternative forced-choice response indicating the perceived right or left direction of motion by pressing the left or right arrows on a keyboard, respectively. The inter-trial interval was fixed at 500 ms. At the beginning of each trial, a central dot flickered on the screen (50 ms on, 50 ms off, 250 ms on, 150 ms off) followed by the stimulus presentation. The initial stimulus duration was set to 150 ms. The duration was further adjusted depending on the participant's response using two (one for small and one for large stimuli) interleaved one-up, two-down staircases (with 8.3 ms steps) that converged on 71% correct performance. The block continued until both staircases completed 9 reversals. All subjects performed two blocks of the task.

The thresholds were obtained for each block by averaging the presentation times corresponding to the reversals and then averaging between the blocks. The two first reversals of each block were excluded from the averaging. We also excluded the reversals, which with high provability were explained by inattention rather than perceptual limitations. This became obvious when an error was followed by a number (N) of correct responses in a row to the gratings presented for the same or shorter durations. We adopted N = 7, because the probability of having 7 correct responses in a row by chance is very low ($1/2^7$ ~ 0.008).

The thresholds were log10 transformed to normalize the distributions. We then calculated the spatial suppression index (SSI)–a measure that quantifies the deterioration of motion

**Fig 1. Overview of the data collection and analysis pipelines.** (a) Psychophysical experiment. Large or small gratings moving left or right were presented for a short period of time. The subject had to indicate the direction of motion by pressing the corresponding button. In case of a wrong response the exposure time increased, while in case of two correct responses in a row it decreased ('one-up—two-down staircase'). The minimum time required for the subject to detect direction of motion (i.e. duration threshold) of small and large gratings was defined as the average exposure time at all but the first two 'reversals' of the corresponding staircase. (b) MEG experiment. The subject was presented with circular gratings drifting toward the center at different speeds. Brain activity corresponding to baseline and visual stimulation intervals was localized using LCMV beamforming. Stimulation-related changes in gamma power (gamma response) were calculated as (stimulation-baseline)/baseline. The peak frequency and power of gamma response (GR) were analyzed at 'maximal sources' in the visual cortex. For a detailed description of the experimental procedures and analysis, see the Methods section.

direction discrimination caused by an increasing stimulus size:

$$SSI = \log10(Threshold_{Large}) - \log10(Threshold_{Small})$$

A larger SSI value indicates a greater deterioration in motion perception with increasing stimulus size.

## 2.3. MEG experiment

The experimental procedure used to induce GRs has been described in our previous studies [35, 56] and is summarized below. A graphical overview of the MEG experiment is presented in Fig 1B.

Children viewed a series of large (18 degrees of visual angle, 1.66 cycles per degree) high-contrast annular gratings that drifted to the center of the visual field with one of three velocities: 1.2°/s ('Slow'), 3.6°/s ('Medium'), 6.0°/s ('Fast'), which corresponds to 2, 6, and 10 Hz in temporal frequency, respectively. Each trial began with the appearance of a white fixation cross that stayed in the center of a black screen for 1.2 s and was followed by a circular grating drifting to the center. After moving for 1.2–3 s, the grating stopped, at which point the subjects were instructed to immediately press a button. Directly after the button press (or, in the

absence of a button press, 2 seconds after termination of the grating's motion), a new trial began. In order to reduce fatigue and increase attention, every 2–5 trials were followed by a short (3–6 s) cartoon movie. 90 gratings of each type were presented to the majority of children during three ~7-minutes experimental blocks with a short resting period between the blocks. In 2 children (1 ASD and 1 TD) the number of epochs per condition was less than 90 (range 60–83) and in 11 children (5 NT and 6 ASD) exceeded 90 (range 105–177). The stimuli presentations were controlled by Presentation software (Neurobehavioral Systems Inc.) and were presented by a PT-D7700E-K DLP projector (screen resolution: 12801024; refresh rate: 60 Hz). During the stimulation condition, the luminance of the screen at the level of the subjects' eyes was 53 Lux; during inter-stimulus intervals, it was 2.5 Lux.

## 2.4. MEG recording and preprocessing

The MEG data recording was performed at the Moscow MEG Center (Moscow State University of Psychology and Education) with an Elekta VectorView 306-channel detector array (Neuromag, Finland) comprising 204 orthogonal planar gradiometers and 102 magnetometers in 102 locations above the participant's head. The MEG signals were acquired with a band-pass filter of 0.03–330 Hz and 1000 Hz sampling rate.

MaxFilter software (v.2.2) was used for initial data preprocessing. The data were de-noised using the temporal Signal-Space Separation (tSSS) method [59] with correlation limit 0.9 and motion corrected. We converted all the data to the subject's 'optimal' initial head position of one of his experimental runs. To select the 'optimal' run, we computed the average position shifts produced by converting the data to each of the initial positions, and chose the one that yielded the smallest average shift across all experimental epochs (-1 to 1.2 s with respect to the stimulus onset) from all runs combined.

Epochs, in which the head position shifted by more than 20 mm after motion correction were further excluded from the analysis. The latter was the case in 3 boys with ASD (15, 20 and 73 epochs) and in 3 control participants (11, 29 and 44 epochs). For the remaining epochs, the deviation from the initial head position did not differ between TD and ASD groups (TD: mean = 3.82 mm, SD = 1.95 mm; ASD: mean = 4.63 mm, SD = 2.57 mm; $T_{(77)}$ = 1.56, p = 0.12).

All further steps of analysis were performed using the MNE-python toolbox (v.0.19) [60]. For each subject, the MEG signal was resampled at 500 Hz and then the experimental blocks were combined into a single file. Suppression of biological artifacts was performed with the Signal Space Projection method (SSP; [61]). For each participant, we excluded one EOG and one ECG projection, which were automatically specified for subtraction. The signals from the 204 planar gradiometers were used for further analysis. The data were band-pass filtered at 40–120 Hz, notch-filtered at 50 and 100 Hz, and epoched with respect to the stimulus onset (-1 to 1.2 s). The epochs contaminated by bursts of myogenic activity or high-amplitude artifacts were marked based on visual inspection of unfiltered data and excluded from the analysis. The minimum number of artifact-free epochs per condition was 40. On average, the number of artifact-free epochs in the TD vs. ASD groups was 70.4 vs. 59.2 for the 'slow', 70.3 vs. 56.9 for the 'medium', and 70.3 vs. 60.3 for the 'fast' condition. To reduce possible contribution of 60 Hz signal entrainment associated with the projector refresh rate, the average evoked response has been subtracted from each data epoch using MNE-python function 'subtract_evoked'.

## 2.5. Structural MRI

Structural MR scans (voxel size 1 x 1 x 1 mm) were performed on a General Electric Signa 1.5 T scanner. The obtained T1 images were processed using default FreeSurfer (v.6.0.0) 'recon-

all' reconstruction algorithm [62]. The results of this reconstruction were used to create the source model as described below.

## 2.6. MEG data analysis

MEG data were co-registered with each subject's structural MRI using the mne_analyze tool (MNE-C software [63]). Single layer BEM models were created using the FreeSurfer watershed algorithm [64]. Next, the forward model was estimated using the surface-based source space with 4096 sources for each hemisphere. Source localization of gamma activity was performed using the Linearly Constrained Minimum Variance (LCMV) beamformer inverse solution algorithm [65] and limited to the visual and adjacent areas that included pericalcarine gyrus, lateral occipital gyrus, cuneus, precuneus, lingual gyrus, fusiform gyrus, inferior and superior parietal gyri, and bank of the superior temporal gyrus [66]. To compute LCMV spatial filters, we used the MNE-python function 'make_lcmv', which takes both noise and data covariance as inputs. We used the same spatial filter for the baseline (-0.9–0 s relative to the stimulation onset) and stimulation (0.3–1.2 s) time intervals across the three motion velocity conditions. The early post-onset interval (0–0.3 s) that contains onset evoked response [67] was excluded from the calculation of the induced GR. The acquired spatial filter was applied to each epoch with the regularization parameter set to 0.05.

For spectral analysis, we used a multi-taper method with 10 Hz bandwidth and frequency resolution ~1 Hz. Time-frequency analysis was performed separately for each vertex source, stimulus condition ('slow', 'medium', 'fast'), and time interval (baseline and stimulation). The normalized spectral power was estimated as [stimulation—baseline] / baseline.

For each stimulus condition we then selected 26 vertices with a maximal stimulus-related increase of spectral power in the 50–80 Hz range. The 'maximal condition' was then defined as the one with the maximal stimulation-related increase in the averaged power. The individually chosen 26 vertices of the 'maximal condition' (usually the 'slow' one) were then used as the regions of interest (ROIs) for the GR analysis (i.e., one ROI per individual across all conditions).

For each of the three velocity conditions, we then obtained the averaged normalized power spectra in the 50–105 Hz range in each subject's ROI. The spectra were smoothed with a 3-point moving average window. Next, we calculated the weighted GR power as an average over those data points that exceeded 2/3 of the peak GR power in the 50–105 Hz range (hereaf-ter–GR power). A weighted GR frequency was estimated as the center of gravity over those frequencies that were used for the GR power estimation (hereafter–GR frequency).

The GR frequency was estimated only in case of reliable GR. To calculate the probability of a stimulation-related increase in power, we applied the Wilcoxon rank test to compare the single trial power in baseline and stimulation intervals at the peak frequency of GR. The GR was considered reliable at p's<0.0001. This strict selection threshold has been adopted here and in our previous studies (e.g. [40]), because noisy GRs would result in unreliable assessment of their state-related changes. Besides, noisy GRs would provide an unreliable estimation of the peak gamma frequency [68]. The individual subjects' GR spectra with highlighted 'reliable' peaks are shown in the *S1 Fig*.

To find brain coordinates of the maximal GR, the individual MRIs were morphed to the FreeSurfer template brain, and the coordinates were estimated in the MNI coordinate system.

To quantify the attenuation of the GR power with increasing motion velocity of the visual grating, we have previously computed the coefficient of a linear regression of the GR power vs. velocity–'gamma suppression slope' [40]. This slope is proportionally more negative with stronger velocity-related attenuation of the GR. In the present study, however, a few children

with ASD demonstrated bell-shaped changes in the GR power with increasing velocity (i.e., GR power increased from the 'slow' to 'medium' velocity condition and then decreased to the 'fast' condition). This led to a strongly non-Gaussian distribution of the slope values in the ASD group. Therefore, to quantify changes in the GR power across different velocity conditions we used gamma response suppression index (GRSI) computed as the center of gravity of the GR power (POW) as a function of visual motion velocity:

$$\mathrm{GRSI} = (\mathrm{POW}_{slow} * 1.2 + \mathrm{POW}_{medium} * 3.6 + \mathrm{POW}_{fast} * 6.0)/(\mathrm{POW}_{slow} + \mathrm{POW}_{medium} + \mathrm{POW}_{fast})$$

The similar index, but based on four velocity conditions (0, 1.2, 3.6, 6.0˚/s), was used in our previous study and appeared sensitive to the level of excitatory drive to the visual cortex [38]. Low GRSI indicates strong suppression of GR power with increasing velocity, while high GRSI indicates weak suppression.

The reliable GR measured in the 'maximal' (usually 'slow' velocity) condition may decrease and become unreliable with increasing velocity of visual motion. Yet, since the estimation of the GR power was based on the analysis performed on the same cortical sources, the responses elicited in different velocity conditions can be meaningfully compared. Therefore, GRSI was estimated if the subject had a reliable GR in at least one of the experimental conditions. Distribution of GRSI values was not significantly different from normal in either the TD or ASD groups (c.f., Results).

## 2.7. Statistical analysis

The Shapiro-Wilk test was used to test for normality of the distributions of all the experimental variables. GR power values were log10—transformed in order to normalize the distributions. To estimate group differences in source coordinates, reaction time (RT), etc., the Student's t-test or Mann-Whitney U test were used in case of Gaussian and non-Gaussian distributions, respectively.

To test for group differences in the effect of age on GR frequency and power, we applied the Homogeneity-of-Slopes analysis with the categorical factor Group and Age as a continuous factor separately in each velocity condition. Age was then used as a covariate in analyses of covariance (ANCOVAs), which have been performed separately for the GR frequency and power in each experimental condition. The effect of grating's velocity on MEG variables was estimated using the repeated measures analysis of variance (ANOVA) with factors Group (TD, ASD), Velocity ('slow', 'medium', 'fast'), and centered Age as a covariate. The Greenhouse-Geisser correction was applied to correct for violation of sphericity assumption. In case of significant repeated measures ANOVA effects, post-hoc planned comparisons (paired t-tests) were used to test for group differences in GR power and/or frequency. The Bartlett Chi-squared test was applied to test for the homogeneity of variance of the GRSI measure in the TD and ASD samples.

To examine the group differences in the motion discrimination thresholds and SSI, duration thresholds were log10-transformed to normalize their distributions and a two-way ANOVA with factors Group and Stimulus Size ('small', 'large') was performed. In addition, in order to evaluate the suppression effect specifically related to the large stimulus size, we estimated residual values of the duration thresholds for large stimulus after partialling out thresholds for the small stimulus using linear regression analysis, as was done in our previous study [17].

Spearman rank correlation coefficients were estimated to measure the degree of association between variables.

## 3. Results

The focus of the present study was the relationship between stimulation-related modulations of MEG gamma oscillations and perception of visual motion in children with ASD. For some children with ASD, it is difficult to tolerate MEG and MRI data collection procedures. Their increased motility and/or poorer attention to visual stimuli may lead to low signal to noise ratio and affect the power and localization accuracy of the visually induced GRs. In addition, the developmental courses of GR parameters can differ in TD and ASD, as has been shown, for example, for alpha rhythm [69], leading to inconsistent results across the age groups.

To make sure that our results were not explained by nuisance factors potentially confounding GR measurements in clinical populations, we estimated the group differences in subjects' head positions during the MEG experiment (see the Methods), in behavioral task performance during MEG recording (section 3.1) and in localization of visual GRs sources (section 3.2). Then, considering the debates in the literature regarding visual GR parameters in a single experimental condition as possible 'biomarkers' of autism [32], we checked for the group differences in GR power and frequency and developmental timecourses of the GR parameters in each stimulation condition separately (section 3.3). Finally, we tested our main prediction regarding the relatively reduced *modulation* of GR power by increasing visual input intensity in the ASD sample (section 3.4) and the correlation of this putative index of weakened neural inhibition in V1 with the psychophysiological thresholds measured outside the MEG scanner (sections 3.5 and 3.6), and tested the contribution of IQ and autism severity to this correlation (section 3.7).

### 3.1. Behavioral performance during MEG experiment

In order to maintain participants' attention to the visual display during the MEG recording, we asked them to press a button when the grating stopped moving. The percent of omission and commission errors in this task was negligible and did not differ significantly between the two groups (median % of omission errors: 0.4 vs. 1.1, commission errors: 2.6 vs. 3.8 for TD and ASD, respectively; Mann-Whitney U test p's>0.05). This means that all participants, including children with ASD, demonstrated a near perfect level of response performance. RTs were significantly longer in children with ASD in all velocity conditions (median RT for 'slow': 431 vs. 579 ms, 'medium': 403 vs. 555 ms, 'fast': 396 vs. 532 ms for TD and ASD, respectively; Mann-Whitney U test, p's<1e-6). The general slowing of simple motor responses in people with ASD is a frequent finding that can be attributed to a variety of factors that are not related to attention, e.g., reduced IQ, clumsiness, and/or delays in motor development [70]. Indeed, in our participants with ASD, RT correlated with MPI IQ (N = 33; 'slow': R = -0.36, p = 0.04; 'medium': R = -0.39, p = 0.02; 'fast': R = -0.41, p = 0.02). In order to ensure that only attended trials were analyzed, we excluded those trials in which RT exceeded 1 second (after the termination of visual motion) from the MEG analysis.

### 3.2. Spatial localization of MEG visual gamma responses

To test for group differences in the localization of the GR maximum, we analyzed MNI coordinates of the vertex with maximal stimulus-related increase of 50–80 Hz spectral power in the 'slow' velocity condition, in which the greatest GR was observed in most participants. Only participants with reliable GRs were included in this analysis (see the Methods). The coordinates did not differ between the groups ($N_{TD}$ = 34, $N_{ASD}$ = 35; $X_{TD}$ = 1.4 $X_{ASD}$ = 0.75; $Y_{TD}$ = -94.7, $Y_{ASD}$ = -95.1; $Z_{TD}$ = 2.9, $Z_{ASD}$ = 0.5; Student's t-test, all p's>0.2). In both groups, the averaged MNI coordinates of the 'maximally induced' vertex corresponded to the right V1 (Bioimage Suite, http://www.bioimagesuite.org; Fig 2). This position corresponds well with the

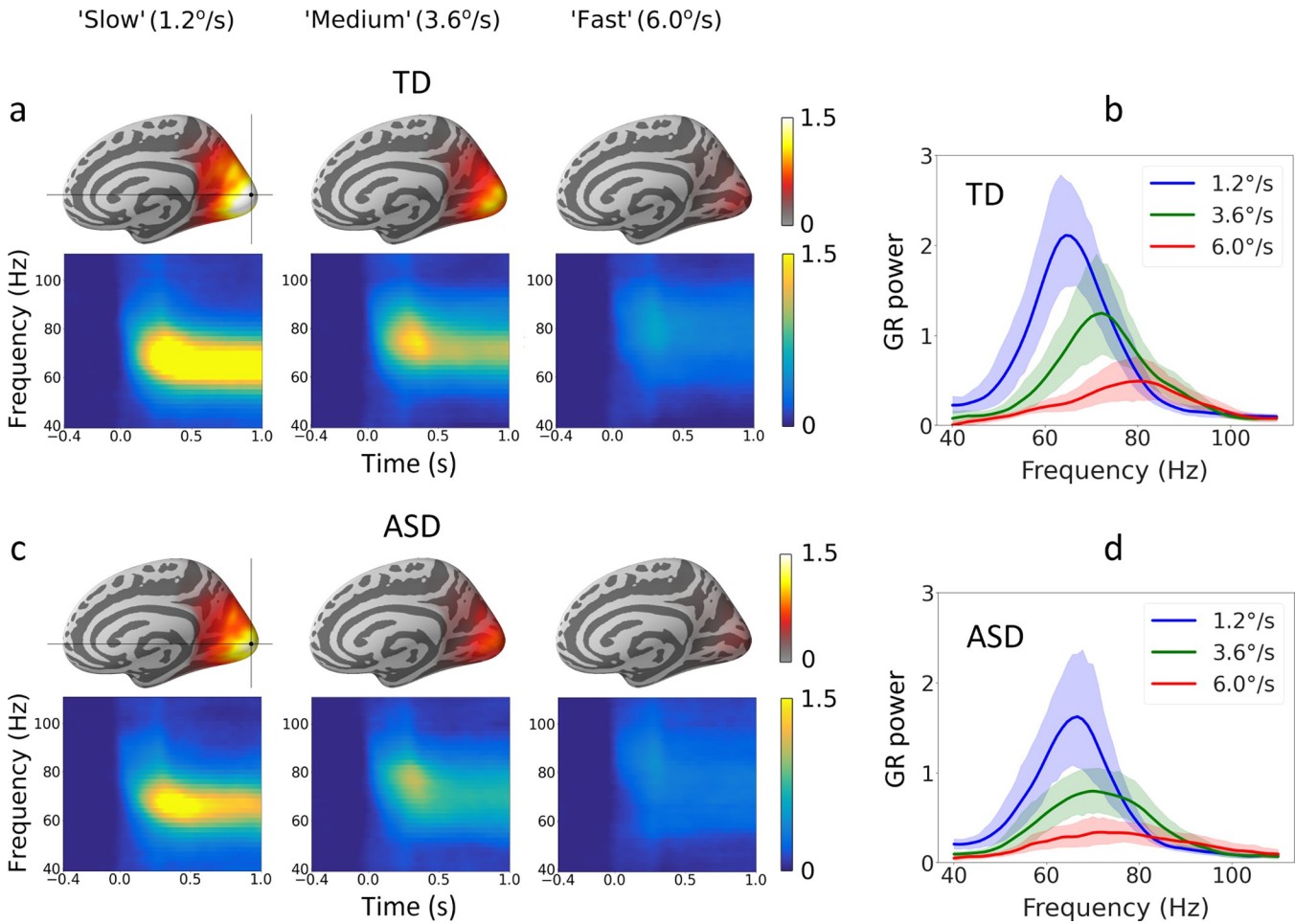

**Fig 2. Group averages of gamma responses (GRs) induced by drifting visual gratings in TD and ASD children.** (a, c) Cortical localization and time-frequency plots of the induced GR power ([stimulation-baseline]/baseline) in each of the three velocity conditions ('slow': 1.2˚/s, 'medium': 3.6˚/s, and 'fast': 6.0˚/s) in TD and ASD participants. Time-frequency plots of the GRs are shown for the average of 26 'maximal' vertices. (b, d) Group average spectra and 95% confidence intervals (shaded areas) for the three velocity conditions.

results of other MEG studies that used beamformer spatial filters for source localization and found the maximum of visual induced GR in the calcarine fissure or surrounding visual cortex [71–74]. It also agrees well with the results of animal studies showing that gamma power increments are generally stronger in area V1 as compared to higher visual areas [75]. There were no group differences in the variance of X, Y, or Z coordinates (Bartlett Chi-squared test, all p's>0.05).

## 3.3. Comparison of MEG visual gamma responses in the TD and ASD groups under the three velocity conditions

**3.3.1. Group differences and maturational trajectories of gamma response power.** All subsequent analyses of the GR were performed at the level of cortical sources (see Methods for details). The distributions of GR power values were strongly non-Gaussian in both the TD and ASD groups (Shapiro-Wilk test; all W's<0.89, p's<0.002). Violin plots of original GR power values for the three velocity conditions and the two experimental groups are shown in S2 Fig. To normalize the distribution, we applied the log10 transformation.

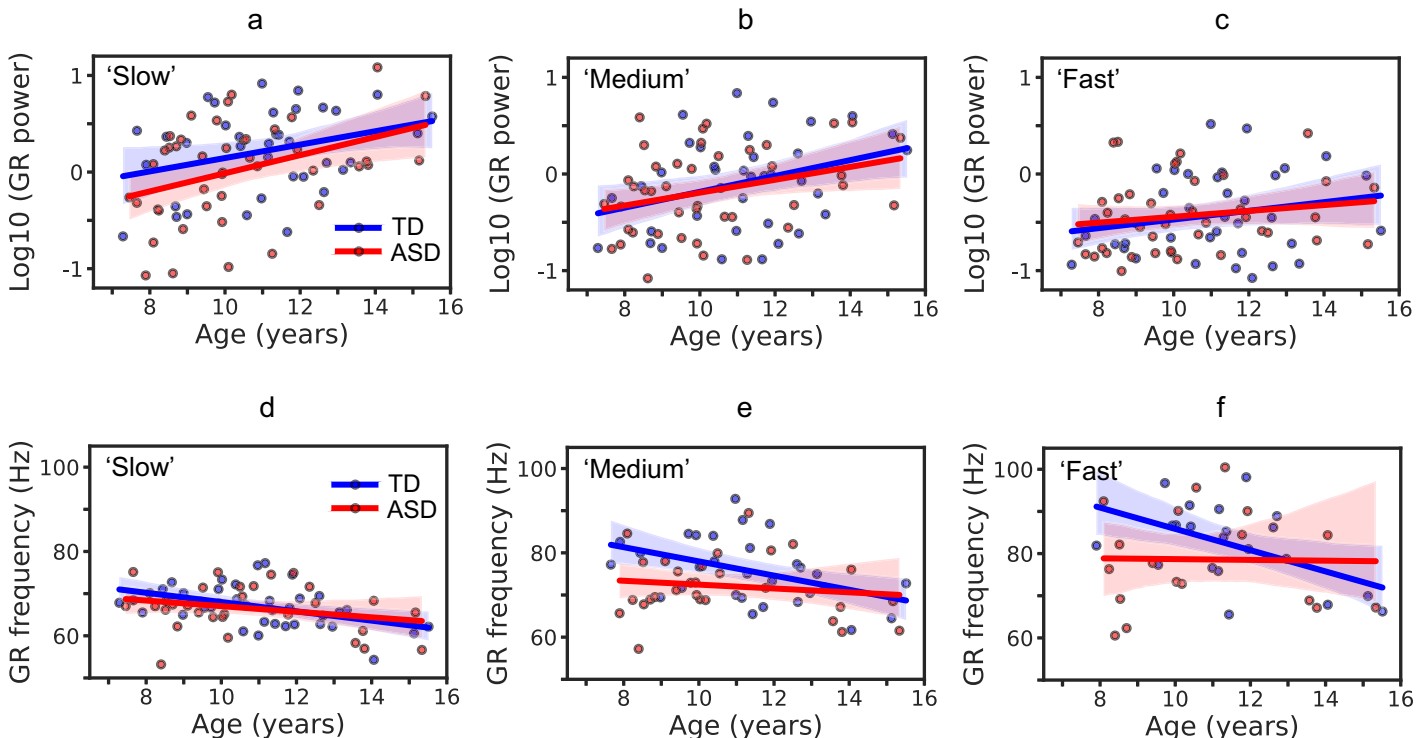

**Fig 3.** Developmental changes in power (a,b,c) and frequency (d,e,f) of gamma responses (GRs) under 'slow' (1.2°/s), 'medium' (3.6°/s), and 'fast' (6.0°/s) velocity conditions in children with ASD and in TD children. Solid lines show linear fit and the shaded areas correspond to 95% confidence intervals.

First, we tested for group differences in the effect of age on GR power using Homogeneity-of-Slopes analysis with the main factor Group and Age as a continuous predictor. There was no Group x Age interaction in any of the velocity conditions (all p's>0.6), suggesting similar developmental trends for the GR power in the two groups. Therefore, we further used age as a covariate in ANCOVA in order to test for the effects of Group and Age.

Separate ANCOVAs were performed for each velocity condition. The GR power increased with age in the case of 'slow' ($F_{(1,76)}$ = 10.80, p = 0.002, partial eta-squared = 0.12) and 'medium' ($F_{(1,76)}$ = 8.85, p = 0.004, partial eta-squared = 0.10) velocity conditions (Fig 3A and 3B). In the case of 'fast' velocity, the effect of Age did not reach significance level ($F_{(1,76)}$ = 2.74, p = 0.10, partial eta-squared = 0.03; Fig 3C). The Group effect was not significant for any velocity condition (Table 3). The variance of the log10-transformed GR power did not differ between the TD and ASD groups (Bartlett Chi-squared test; p's>0.29 for all conditions).

Thus, the power of visual GR increased with age in both TD children and children with ASD, and did not differ between the groups in any particular velocity condition.

**3.3.2. Group differences and maturational trajectories of gamma response frequency.** We analyzed the frequency of GR only in case of a reliable stimulus-related increase in gamma power. The number of subjects who fulfilled the criterion for the GR reliability decreased from the 'slow' to 'fast' velocity condition ('slow': $N_{TD}$ = 35, $N_{ASD}$ = 34; 'medium': $N_{TD}$ = 29, $N_{ASD}$ = 30; 'fast': $N_{TD}$ = 21, $N_{ASD}$ = 22). The distributions of the GR frequency values did not differ from normal in either the TD or ASD groups (Shapiro-Wilk test; all W's>0.95, p's>0.15 for all groups and conditions).

For the frequency of the GR, Homogeneity-of-Slopes analysis did not reveal statistically significant group differences in the age trends (Age x Group effect for 'slow': $F_{(1,65)}$ = 0.57,

**Table 3. Power of gamma response (GR) in TD and ASD groups of children and analysis of group differences (ANCOVA results).**

| | TD* median (range) | ASD* median (range) | $N_{TD}/N_{ASD}$ | F** | p*** | Partial eta² |
|---|---|---|---|---|---|---|
| Slow (1.2˚/s) | 2.00 (0.22–8.24) | 1.25 (0.09–12.12) | 37/42 | 1.84 | 0.17 | 0.024 |
| Medium (3.6˚/s) | 0.94 (0.13–6.88) | 0.70 (0.08–3.86) | 37/42 | 0.04 | 0.84 | 0.001 |
| Fast (6.0˚/s) | 0.31 (0.08–3.29) | 0.33 (0.10–2.64) | 37/42 | 0.04 | 0.85 | 0.000 |

*Power of GR is expressed as a change: [stimulation—baseline] / baseline

** F test was performed using log10-transformed GR power values

*** Uncorrected for multiple comparisons

p = 0.45, partial eta-squared = 0.01; 'medium': $F_{(1,55)}$ = 2.02, p = 0.16, partial eta-squared = 0.04; 'fast': $F_{(1,39)}$ = 2.28, p = 0.14, partial eta-squared = 0.06). Therefore, we further used age as a covariate in the ANCOVA.

As in the case of the GR power, separate ANCOVAs on GR frequency with factors Group and Age were performed for each velocity condition. The frequency decreases with age in the 'slow' ($F_{(1,66)}$ = 8.95, p = 0.004, partial eta-squared = 0.12) and 'medium' ($F_{(1,56)}$ = 4.25, p = 0.044, partial eta-squared = 0.07) velocity conditions, but the effect of age did not reach significance level in the 'fast' velocity condition ($F_{(1,40)}$ = 2.14, p = 0.15, partial eta-squared = 0.05). When each group was analyzed separately, the TD group demonstrated a significant age-related decrease in GR frequency under each of the three velocity conditions ('slow': $F_{(1,29)}$ = 9.33 p = 0.005; 'medium': $F_{(1,29)}$ = 6.61, p = 0.02; 'fast': $F_{(1,29)}$ = 5.33 p = 0.03). Such a developmental decrease in the GR frequency was virtually absent in the ASD group ('slow': $F_{(1,23)}$ = 0.09, p = 0.76; 'medium': $F_{(1,23)}$ = 0.07, p = 0.78; 'fast': $F_{(1,23)}$ = 0.19 p = 0.66). The group differences in GR frequency are summarized in Table 4.

To summarize, the GR was localized to the V1, and neither its mean X, Y, and Z coordinates nor their variability differed between the TD and ASD groups. GR power in V1 did not differ between the TD and ASD samples in any particular motion velocity condition. The GR frequency during the 'medium' condition was lower in children with ASD, but the difference did not survive correction for multiple comparisons. Maturational changes led to an increase in

**Table 4. Frequency of gamma response (GR) in TD and ASD groups of children and analysis of group differences (ANCOVA results).**

| | TD mean ± SD (range) | ASD mean ± SD (range) | $N_{TD}/N_{ASD}$* | F | p** | Partial eta² |
|---|---|---|---|---|---|---|
| Slow (1.2˚/s) | 66.8 ± 5.0 (54.3–77.2) | 66.6 ± 5.4 (53.2–75.1) | 35/34 | 0.24 | 0.62 | 0.004 |
| Medium (3.6˚/s) | 75.7 ± 7.6 (61.7–92.8) | 72.1 ± 7.2 (57.2–89.4) | 29/30 | 5.17 | 0.03 | 0.085 |
| Fast (6.0˚/s) | 82.3 ± 9.3 (65.5–98.1) | 79.3 ± 10.9 (60.5–100.5) | 21/22 | 1.50 | 0.22 | 0.036 |

* Number of participants with reliable GR

** Uncorrected for multiple comparisons

the GR power and a decrease in GR frequency with age, although the age-related decrease in GR frequency was not significant when analyzed separately in the ASD sample.

### 3.4. Modulation of the MEG gamma response power and frequency by stimulus velocity in TD and ASD

To test for the effect of visual motion velocity on the GR power and frequency and its interaction with group, we applied repeated measures ANOVA with factors Group and Velocity.

**3.4.1. Velocity-dependent suppression of the MEG gamma response power.** In accord with our previous findings in children and adults [35], we found a strong main effect of Velocity on GR power ($F_{(2, 152)}$ = 182.5, epsilon = 0.72, adjusted p<1e-6, partial eta-squared = 0.71), i.e., GR power attenuated with increasing drift rate of the grating. A significant Velocity x Group interaction ($F_{(2, 152)}$ = 4.0, epsilon = 0.72, adjusted p = 0.03, partial eta-squared = 0.05) was explained by a sharper drop in the GR power with increasing velocity in the TD as compared to ASD sample (Fig 4A).

**3.4.2. Velocity-dependent increase of the MEG gamma response frequency.** Consistent with our previous results [35] and those of other research groups [36, 76], we observed a strong and highly significant increase in GR frequency with increasing velocity of visual motion in the combined TD and ASD sample ($F_{(2, 80)}$ = 149, epsilon = 0.70, adjusted p<1e-6, partial eta-squared = 0.79). The GR frequency increased, on average, from 66 Hz at 'slow' velocity to 81 Hz at 'fast'. Of note, the number of subjects who were included in the GR frequency ANOVA analysis was approximately twice as small as that for the corresponding analysis of the GR power (21 out of 37 TD and 22 of 42 ASD children; Table 4). The remaining subjects provided unreliable information regarding GR frequency at the 'fast' velocity because of a low signal to noise ratio (see Methods for details). Therefore, to increase the statistical power of the between-group comparisons, we repeated the ANOVA including only 'slow' and 'medium' velocity conditions, wherein reliable GRs were recorded for the majority of the subjects (29 of 37 TD and 29 of 42 ASD, respectively; Table 4). There was a significant Group x Velocity

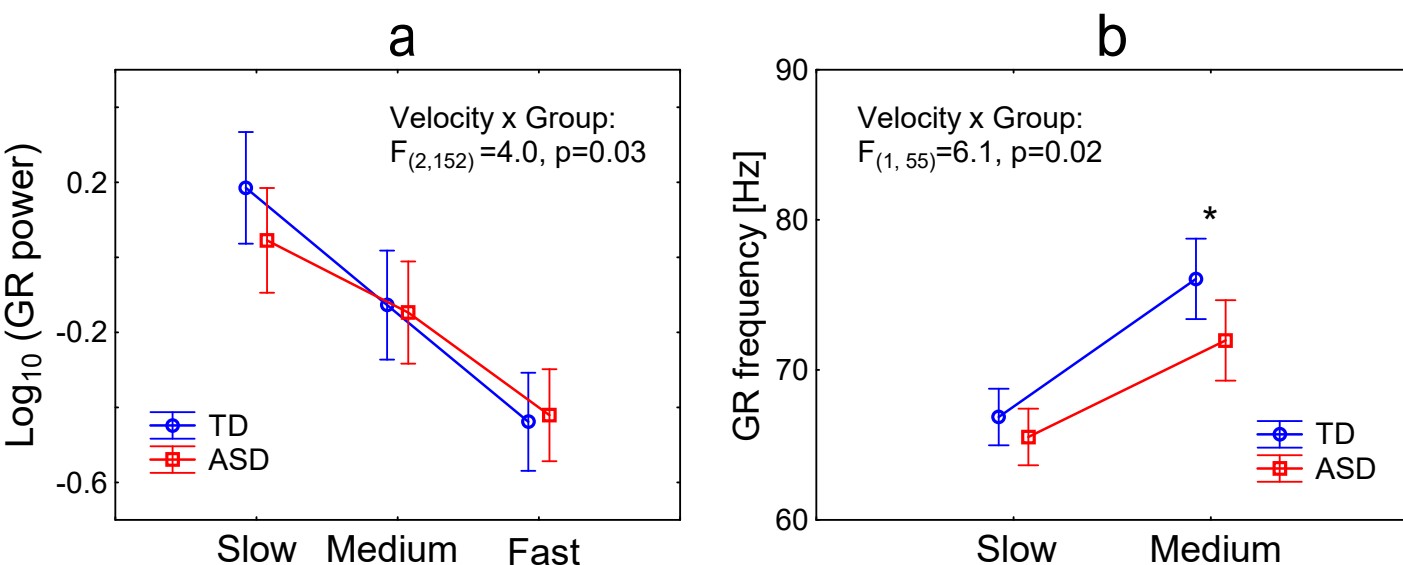

**Fig 4.** Velocity-dependent changes in gamma response (GR) power (a) and frequency (b) in TD and ASD children. The main figure in panel 'b' shows the results for children with reliable GRs in both the 'slow' and 'medium' conditions ($N_{TD/ASD}$ = 29/30). *p = 0.036, uncorrected for multiple comparisons.

interaction ($F_{(1, 55)}$ = 6.1, p = 0.02, partial eta-squared = 0.1, Fig 4B): the increase in GR frequency from the 'slow' to 'medium' velocity condition was reduced in the ASD, as compared to the TD, group.

To summarize, modulations of the GR power and frequency by increasing the grating's drift rate were reduced in the visual cortex of children with ASD at the group level.

**3.4.3. Inter-individual variability in the gamma response suppression in children with ASD.** Considering the heterogeneity of the ASD population, the differences in GR suppression may characterize only some individuals. Therefore, as a next step we estimated interindividual variability of the GR suppression.

The GR suppression can be estimated by calculating GRSI (Fig 5A), but only in those children who have reliable GR in at least one condition (usually it was the 'slow' velocity condition). This was the case in 34 of 37 TD and 34 of 42 ASD children. The participants who had unreliable GRs in all experimental conditions (3 TD and 8 ASD) were significantly younger than the rest of the sample (Age 9.1 years vs. 10.9 years; Mann-Whitney U test: U = 183, Z = 2.7, p = 0.007, effect size r = 0.3). The ASD children with unreliable GRs did not differ from the rest of the ASD sample in MPI IQ (mean$_{unreliable}$ = 86.6, mean$_{reliable}$ = 84.1; Mann-Whitney U test: U = 131, Z = -0.14, p = 0.45, r = 0.01) or SRS scores (mean$_{unreliable}$ = 100.4, mean$_{reliable}$ = 110.1; Mann-Whitney U test: U = 93, Z = 0.95, p = 0.34, r = 0.1).

Variability of GRSI was higher in the ASD than in the TD group (Bartlett Chi-squared = 4.62, p = 0.03), but its mean value did not differ significantly between children with ASD and TD participants ($T_{(1,66)}$ = 1.8, p = 0.07, partial eta-squared = 0.05). The violin plots of the GRSI values are presented in Fig 5B and were not significantly different from Gaussian in either TD (Shapiro-Wilk W = 0.97, p = 0.48) or ASD (Shapiro-Wilk W = 0.94, p = 0.07) groups, although it was skewed to high GRSI values in the ASD group, with some children having very high GRSI. In three out of thirty-four ASD participants for whom GRSI was calculated, it was in the outliers range according to the 'measure of spread' test [77]. In contrast to all other children, these participants did not suppress the GR with transition from the 'slow' to the 'fast' visual motion velocity.

Fig 6 shows GR spectra of children with the five highest (a-j) and five lowest (k-t) GRSI values from each group. Considering the high magnitude of GRs in the 'GRSI outliers' (Fig 6; greater than 200% increase in GR power in the 'maximal GR condition' relative to prestimulus baseline), it is clear that their reduced GR suppression is not an epiphenomenon of a low signal to noise ratio, but reflects atypical modulation of visual GR by the drift rate of the grating. Compared to other ASD participants, the 'GRSI outliers' had the highest GR power in the 'fast' than in the 'slow' velocity condition (see in Fig 6A–6C).

## 3.5. Directional sensitivity to visual motion: Results of psychophysical experiment

Both the psychophysical data and reliable GRSI were available in 31 TD and 26 ASD participants (see Table 2 for their demographic characteristics). The discrimination thresholds for either Small or Large gratings did not correlate with age or MPI IQ in ASD and TD groups (all p's>0.2).

To test whether the previously reported group differences in motion direction discrimination thresholds [17] were present in this smaller sample of participants, we performed ANOVA with factors Group (TD, ASD) and Size (Large, Small). The main effect of Size was highly significant ($F_{(1,55)}$ = 85.3, p<1e-6, partial eta-squared = 0.61; Fig 7) and planned comparison tests confirmed the presence of spatial suppression in both TD ($F_{(1,55)}$ = 66.7, p<1e-6) and ASD ($F_{(1,50)}$ = 25.4, p = 0.000005) participants.

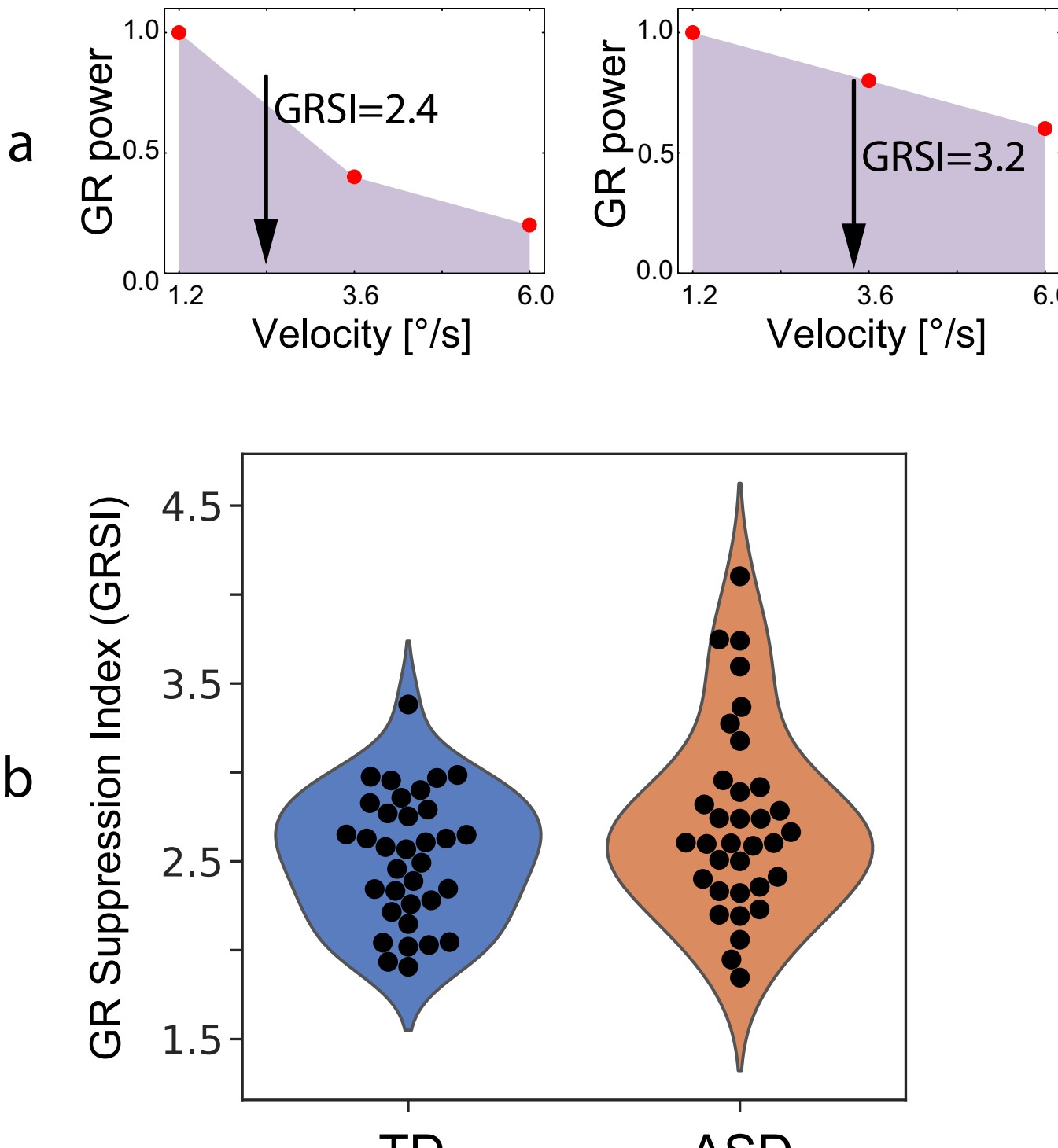

**Fig 5. Gamma response suppression index (GRSI).** (a) Schematic representation of the GRSI calculation. GRSI corresponds to the center of gravity indicated by an arrow: left/right panels show cases of strong/weak gamma response (GR) suppression, respectively. Higher GRSI value indicates weaker GR suppression with increasing visual motion velocity. (b) Individual variability of GRSI in the TD and ASD children: a violin-plot.

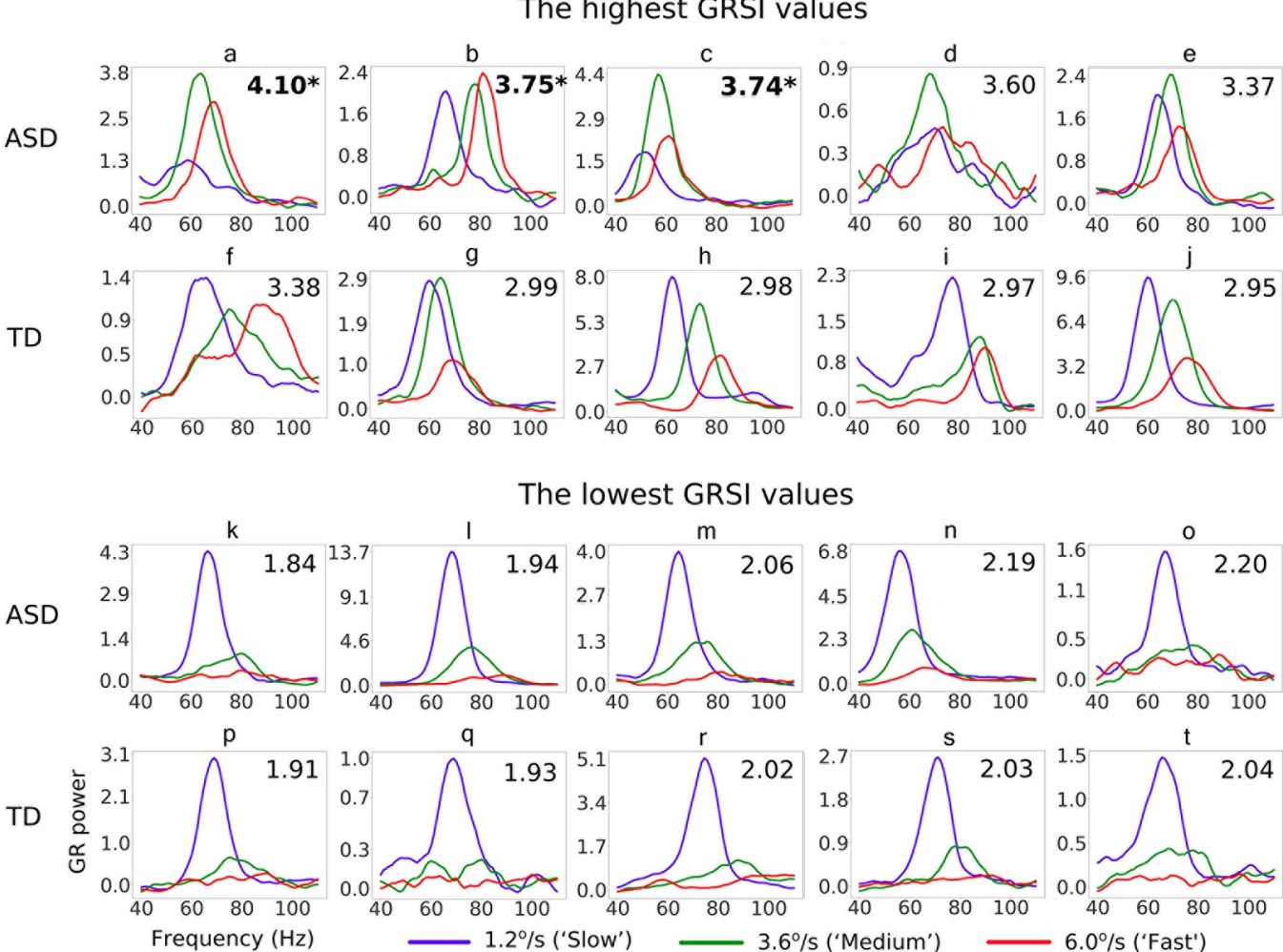

**Fig 6.** Velocity-related modulation of gamma response (GR) spectrum in TD and ASD children with the five highest (upper panels a-j) and five lowest (lower panels k-t) GRSI values. The GRSI values are shown in each individual's plot (top right corner); the GRSI values in the outliers' range are highlighted in bold. Relatively higher GRSI values indicate a weaker suppression of GR by increasing motion velocity.

In contrast to our previous study with a larger number of participants [17], the TD vs. ASD differences in spatial suppression did not reach a level of significance, as evidenced by the Size by Group interaction ($F_{(1,55)}$ = 3.2, p = 0.08, partial eta-squared = 0.06) or group differences in SSI ($T_{(55)}$ = 1.8, p = 0.08). However, duration thresholds for the small grating were higher in the ASD group than in the TD group ($F_{(1,55)}$ = 5.5, p = 0.02, partial eta-squared = 0.09; Fig 7). Thus, the deficit in motion direction sensitivity to small-sized stimuli was still evident in this smaller sample of participants with ASD.

### 3.6. MEG gamma response suppression in V1 correlates with perceptual visual motion sensitivity

First, we tested our main prediction that higher motion duration thresholds for small gratings in children with ASD are associated with weaker GR suppression (higher GRSI). Second, we tested whether children with ASD have the same relationships between spatial suppression (higher motion duration thresholds for large, as compared to small, gratings) and GRSI that we previously observed in TD children [40].

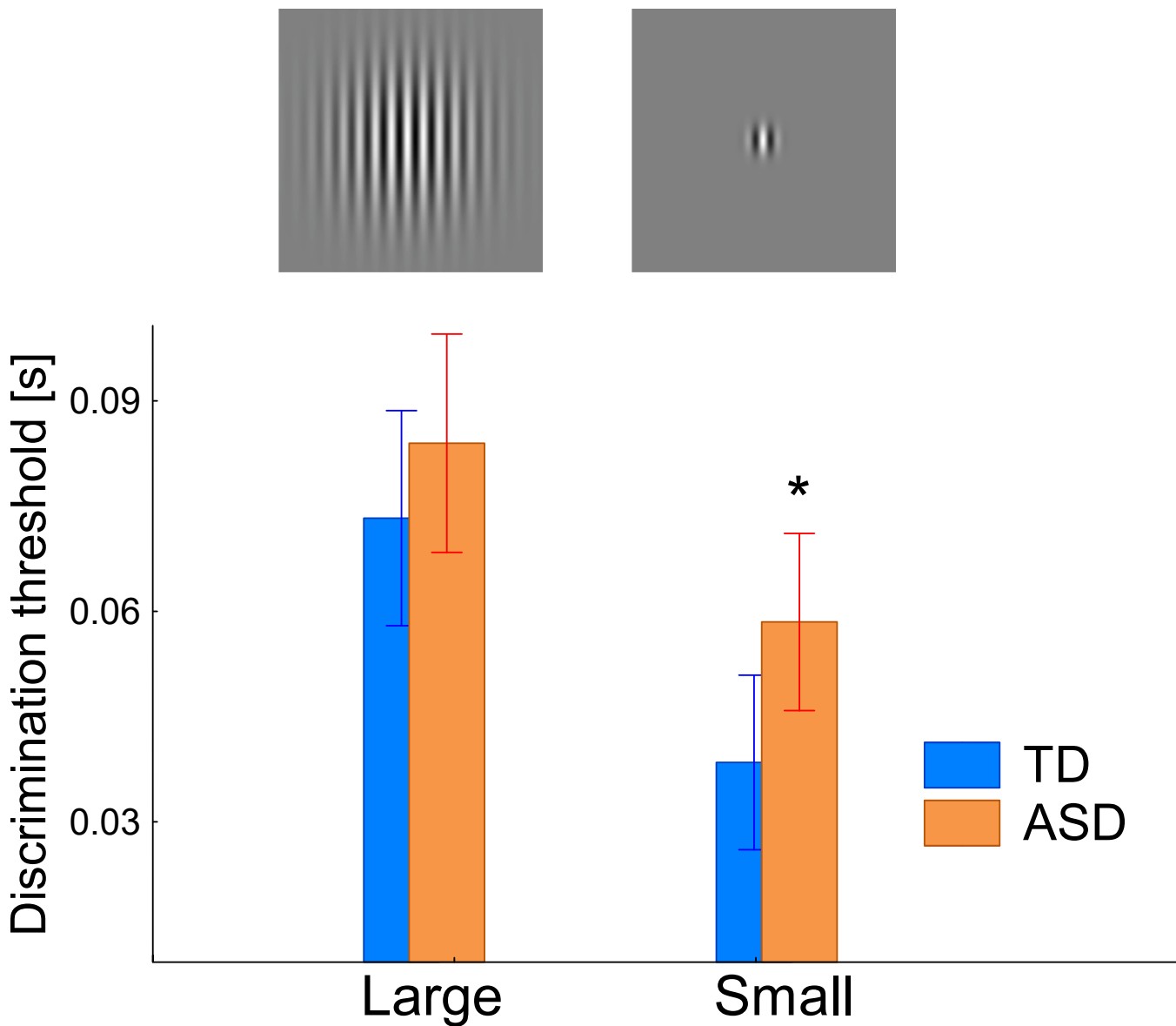

**Fig 7. Motion direction discrimination thresholds for the small and large gratings in children with TD and ASD.** *Between-group difference: p = 0.02.

To test the main prediction, we calculated the Spearman correlations between the duration threshold for small gratings and GRSI. Indeed, in the ASD group, higher motion duration thresholds for small gratings were associated with weaker GR suppression (higher GRSI) (Spearman $R_{(26)}$ = 0.54, p = 0.004; Fig 8A). This correlation remained significant when the GRSI outliers (two of them had psychophysical data) were excluded from the analysis ($R_{(24)}$ = 0.47, p = 0.02). In the TD group, on the other hand, duration thresholds for the small gratings did not correlate with the GRSI ($R_{(31)}$ = 0.12, p = 0.51). The group difference in correlation coefficients was significant (one-tailed Z = 1.72, p = 0.04), confirming that this neurobehavioral relationship was specific to the ASD group. It should be noted that the use of a one-tailed test here is justified, because we had a prediction regarding the direction of the difference, not just its presence [78].

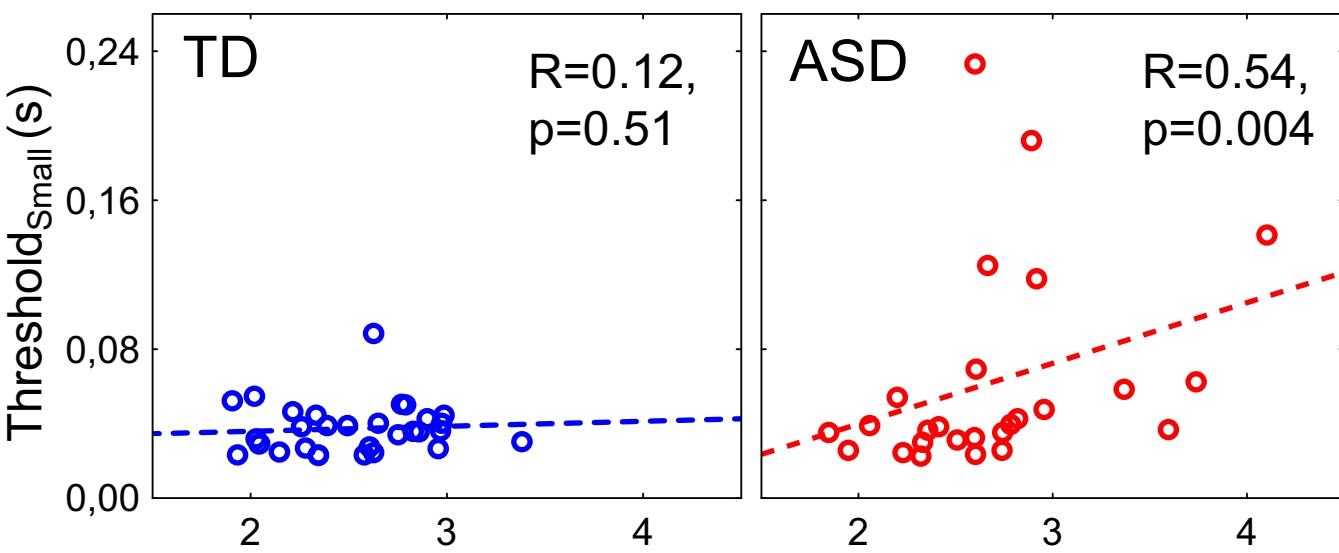

**Fig 8. Spearman correlations between gamma response suppression index (GRSI) and duration threshold for the small grating.** Lower GRSI values indicate stronger velocity-related suppression of the gamma response.

If some children with ASD were generally less attentive than others, this might impair their perceptual discrimination of the small stimuli and, at the same time, affect their GRSI via an overall decrease in the GR power [71]. This 'attentional' explanation assumes negative correlations between perceptual thresholds and GR power. However, the only significant correlation between $Threshold_{Small}$ and GR power was positive (Table 5; for the 'fast' velocity condition: $R_{(26)} = 0.44$, $p = 0.02$, uncorrected). Therefore, the observed ASD-specific correlation between GRSI and sensitivity to the direction of motion for small gratings is unlikely to be caused by inter-individual differences in attention capacity.

Next, we explored the relationship between GR suppression and perceptual spatial suppression. While in TD children SSI correlated with both $Threshold_{Large}$ ($R_{(31)} = 0.62$, $p = 0.0002$) and $Threshold_{Small}$ ($R_{(31)} = -0.51$, $p = 0.004$), in the ASD group's it mainly correlated with duration thresholds for the small gratings (SSI vs. $Threshold_{Large}$: $R_{(26)} = 0.36$, $p = 0.07$; SSI vs. $Threshold_{Small}$: $R_{(26)} = -0.64$, $p = 0.0004$). Therefore, to estimate the effect of spatial suppression specifically associated with increasing stimulus size, we partialled out the effect of the small stimulus, as we did in the previous study [17]. In TD children, this had minimal effect on the correlation between spatial suppression magnitude and GRSI (GRSI vs. SSI: $R_{(31)} = -0.37$, $p = 0.04$; $R_{partial(31)} = -0.35$, $p = 0.054$). In children with ASD, on the other hand, the partial correlation was close to zero (SSI: $R_{(26)} = -0.36$, $p = 0.07$; $R_{(26)} = -0.02$, $p = 0.92$). Thus, the

**Table 5. Spearman correlations between directional sensitivity to the small moving grating ($\log_{10}[Threshold_{Small}]$) and age-corrected gamma response power ($GR_{power}$) in TD and ASD children.**

|  | $\log_{10}[\text{Threshold}_{Small}]$ | |
|---|---|---|
|  | TD (N = 31) | ASD (N = 26) |
| Slow (1.2˚/s) | R = 0.23 | R = 0.09 |
| Medium (3.6˚/s) | R = 0.26 | R = 0.33 |
| Fast (6.0˚/s) | R = 0.26 | **R = 0.44**[*] |

[*]p = 0.02, uncorrected for multiple comparisons

relatively weaker gamma suppression (higher GRSI) had different perceptual concomitants in the ASD and TD groups: it characterized those TD children who were less susceptible to the suppressive effect of increasing stimulus size and those children with ASD who had worse motion sensitivity to the small stimuli.

Overall, our findings showed that children with ASD have an atypical relationship between velocity-related suppression of their visual GR and visual motion sensitivity. In the ASD group, the lower GR suppression in V1 (i.e., higher GRSI) was associated specifically with poor sensitivity to the motion direction of small gratings. In the TD group, on the other hand, the lower GR suppression was associated with better motion direction sensitivity to the large gratings (i.e., weaker perceptual suppression).

### 3.7. Impact of MPI IQ and autism severity on gamma response properties in ASD sample

Given the wide range of intellectual abilities (IQ scores) and autism symptoms (SRS scores) in our participants with ASD, we tested for the impact of these psychometric variables on GR parameters, psychophysical thresholds, and their correlations.

Neither of the GR parameters correlated significantly with the psychometric variables (*S1 Table*). This indicates a relative independence of these GR properties from the severity of communication difficulties or the degree of intellectual deficit in children with ASD aged 7–12 years. The only exception was the GR frequency in the 'slow' and 'medium' velocity conditions, which correlated negatively with IQ. However, these correlations did not survive correction for multiple comparisons.

To assess the contribution of IQ and SRS to the ASD-specific correlation between GRSI and duration thresholds for the small grating, we calculated Spearman partial correlation, while controlling for MPI IQ and SRS. The correlation remained significant ($R_{partial}$ = -0.54, p = 0.007).

## 4. Discussion

We used a putative index of neural inhibition in the visual cortex, the magnitude of velocity-related suppression of visual GR, to assess the contribution of abnormal surround inhibition to previously reported differences in visual motion direction sensitivity in children with ASD. Visual GRs and motion direction sensitivity were assessed in two different experiments (MEG and psychophysical, respectively). Children with ASD demonstrated impaired ability to discriminate the direction of motion of small (1˚) stimuli and weakened GR suppression. Moreover, in the ASD group, there was a correlation between impaired directional sensitivity and weakened GR suppression. We argue below that these findings in children with ASD can be explained by an inhibitory deficit in V1.

### 4.1. Decreased GR suppression and increased duration thresholds indicate a local inhibitory deficit in V1 in children with ASD

The magnitude of the GR induced in V1 by large drifting gratings is determined by the contributions of excitatory and inhibitory activity from the classical and extraclassical receptive fields [79]. The suppression of GR power at high drift rates that we observed in the present (Figs 2B, 2D and 4A) and previous [35, 38, 40–42, 56] studies can be explained by a decrease in the E-I ratio caused by an increase in excitatory drive (see [38] for a detailed discussion). Indeed, computational modeling results predict that GR suppression should occur at a certain high level of excitatory drive due to 'excessive' inhibition, which disrupts gamma synchronization

[39, 80]. Experimental studies in animals do show that a gradual increase in excitatory drive to V1 by increasing visual contrast leads to a greater increase in inhibitory, as compared to excitatory, neurotransmission [34, 81]. In humans, visual contrast and drift rate have an additive effect on GR suppression [38], which is consistent with the predicted desynchronizing effect of strong excitatory drive on visually induced gamma oscillations [39]. Increasing the drift rate of the high-contrast grating beyond a certain value led to suppression of GR also in monkeys, in which it was followed by a decrease in neuronal firing [31]. Overall, these studies suggest a role for neural inhibition in drift-rate-related suppression of visual GR. Unlike GR power, the frequency of visual gamma oscillations, which mainly depends on the activation of inhibitory neurons (INs) [82, 83] increases almost linearly with an increase in the grating drift rate in humans [38] and non-human primates [31]. Thus, the disproportionate increase in inhibition compared to excitation explains why at relatively fast drift rate of high-contrast gratings the GR power decreases and GR peak frequency increases (Fig 4).

Following this line of reasoning, weaker GR suppression at high levels of excitatory drive should reflect less efficient inhibition and/or an elevated E-I ratio. In indirect support of this assumption, we have recently described a lack of velocity-related gamma suppression in a subject with epilepsy [35] and its attenuation in adults with increased sensory sensitivity [41, 42].

Here, we found that frequency and power modulations of the GR under the influence of a strong excitatory influence (fast drift rate) were reduced in children with ASD compared with control TD children (Fig 4A and 4B). While atypically weak GR frequency modulation has been reported previously for a subsample of these children using sensor analysis [56], attenuated GR power suppression is a new finding. While in all TD participants GR decreased from the 'slow' to 'fast' condition, in some autistic children the transition from the 'slow' to the 'fast' visual motion led to an increase in GR power (Fig 6, upper panels 6a-6c). Modeling studies show that an elevated E-I ratio impedes desynchronization of neuronal activity under the influence of strong excitatory input and promotes gamma oscillation [39]. Therefore, the reduced or absent suppression of GR in some children with ASD suggests that surround inhibition recruited by high-intensity visual input ('fast' motion) was not strong enough to desynchronize gamma oscillations. Overall, the decreased modulation of the frequency and power of visual gamma oscillations by strong visual input in children with ASD suggest reduced effectiveness of the 'on-demand' local inhibition in V1.

This inhibitory deficit might be associated with dysfunction of parvalbumin expressing (PV +) interneurons, which play a key role in the generation of gamma rhythms [84], and are strongly implicated in the pathophysiology of ASD. There is a growing consensus in the literature that PV+ INs hypo-function is fundamental to the pathogenesis of autism and, in particular, to the changes in visual function that are often associated with this disorder (for a recent review see [8]). Indeed, the PV+ INs in V1 are hypoactive in many animal models with genetic mutations associated with autism (FRAX, RTT, SHANK3). In Fmr1-knockout mice, this deficiency is causally related to impaired visual discrimination, since artificially increasing PV + neuronal activity restores V1 orientation sensitivity [49]. The causal role of PV+ INs in low-level visual function was also demonstrated by Lee and colleagues, who reported that optogenetic activation of PV+ INs in the V1 sharpened orientation tuning and enhanced direction selectivity of nearby principal neurons in mice [85].

The increase in duration thresholds specifically for small stimuli in children with ASD (Fig 7) is consistent with data on impaired function of PV+ neuron in the primary visual cortex in animal models of autism. Indeed, whereas directional sensitivity to large stimuli is affected by top-down feedback to V1 from higher-tier cortical areas, the directional sensitivity to small (~1˚) gratings depends mainly on the strength of local inhibition in V1 [23, 86]. Of note, our results agree with those of Schauder et al. [16], who linked selective deficit in the ability to

detect motion direction of small gratings in children with ASD to atypical organization of the receptive fields in V1. Surprisingly, two other studies using similar experimental paradigms found atypically *low* duration threshold in people with ASD and average or above-average IQ [18, 19]. To explain such contradictory results, Schallmo et al. proposed a model that explains both increased and decreased directional sensitivity in ASD solely by top-down gain adjustment (e.g. attention allocation or a top-down input from extrastriate areas, such as V5/MT) [19].

The link between GR suppression and duration threshold for the small-size grating in children with ASD observed in our study (Fig 8) is not consistent with such a universal 'top-down' model. Indeed, attention has opposite effects on duration thresholds and GR power [71] and cannot explain presence of positive correlation between GR power and duration threshold for the small grating (Tab. 5). An abnormal top-down modulation of V1 by the higher-tier visual areas also cannot explain the correlation between GRSI and Threshold$_{Small}$ because the strength of such modulation is negligible for small stimuli (~1˚) and increases with increasing stimulus size, as a result of activation of the 'far surround' of the neurons' receptive fields [23, 86]. Thus, the size-dependent deficit of motion perception in children with ASD most provably reflects a deficit in local E-I interactions in the receptive fields of V1 principle neurons and/or their 'near surround' [22].

Insufficient inhibitory influence on V1 principle neurons from their 'near surround' may explain the correlation between GR suppression and directional sensitivity to small gratings in children with ASD. Indeed, since PV+ interneurons sharpen motion direction selectivity in pyramidal cells to which they are connected [50], their dysfunction is expected to impair motion direction discrimination. On the other hand, an inhibitory deficit in V1 is expected to reduce velocity-related suppression of the visual GR, as discussed above. Three our participants with ASD demonstrated the extremely low or no GR suppression, as indicated by their high GRSI values (Fig 6, panels 6a-6c). We previously observed the same atypical lack of velocity-related GR suppression in an adult with epilepsy [35]. This observation is in line with the hypothesis that 'GRSI outliers' have a particularly strong increase in the E-I ratio in the visual cortex. Notably, the atypical correlation between GRSI and Threshold$_{Small}$ in ASD remained significant even when the GRSI outliers (who presumably had an elevated E-I ratio in V1) were excluded from the analysis. The latter suggests that even in the case of a compensated E-I ratio, the E-I interactions in the early visual cortex are still atypical in children with ASD. Of note, the recent MRS results has shown impaired regulation of GABAergic signaling in visual cortex in people with ASD [24], which may contribute to their atypical motion processing.

## 4.2. Different perceptual correlates of GR suppression in TD and ASD

In both groups, increasing the size of the visual grating resulted in an increase in the time required to correctly determine the direction of visual motion (Fig 7). Suppressive feedback from higher-tier visual areas plays a pivotal role in this spatial suppression phenomenon [23, 86] as well as in the synchronization of locally generated gamma oscillations over a large area of visual cortex [29]. The higher-tier visual areas modulate E-I interactions in V1 through potentiation of somatostatin-containing INs [33, 87, 88], which in turn affect the E-I ratio (and gamma oscillations) through the layer-specific inhibition of PV+ and principal cells [89].

Consistent with our previous results [40], GR suppression (quantified as GRSI) correlated with spatial suppression (estimated as SSI or RES$_{Large}$) in TD children. At the same time, no correlation between GR suppression and Threshold$_{Small}$ was found in the TD group. This pattern of correlations suggests that interindividual variability in surround suppression in V1

normally depends on top-down modulation from higher-tier visual areas and is relatively independent of 'good enough' local inhibition in V1.

In children with ASD, on the other hand, the trend toward a correlation between SSI and GRSI was driven by impaired sensitivity to small gratings and completely disappeared when the effect of Threshold$_{Small}$ was partialled out. It is likely that the local inhibitory deficit in the V1, which explains the atypically strong association between GRSI and directional sensitivity to small gratings in children with ASD, obscured the normal relationships between GRSI and spatial suppression.

### 4.3. GR power does not distinguish between TD children and children with ASD

In contrast to GR suppression, neither the GR power measured in separate velocity conditions nor its developmental trends distinguished children with ASD from their TD peers (Figs 3 and 4). The developmental decrease in GR frequency and increase in GR power are compatible with other reports [48, 90] and, as we discussed elsewhere, may reflect fundamental developmental changes in excitatory and inhibitory neurotransmission that are common for children with and without ASD [35]. These 'negative' findings add to the conflicting results of previous attempts to use induced gamma oscillations as a biomarker of altered E-I balance in autism (see [32]). Even if E-I imbalance in V1 affects visually induced gamma oscillations, it is not necessarily reflected in the GR parameters measured in a single experimental condition. Indeed, interindividual variations in cortical anatomy [36, 91], SNR, and other nuisance variables can affect GR power and mask group difference caused by differences in E-I ratio. Moreover, animal studies show that even 'subtle' changes in stimulus features (size, contrast, velocity, orientation, stimulus configuration) engage qualitatively different excitatory-inhibitory interactions in V1 [92, 93], which, in turn, may be differently affected (or not affected) in ASD. In this regard, *changes* in GR during controlled manipulations of targeted stimulus features (e.g., drift rate of a grating in this study) may provide a better framework for approaching neural basis of specific visual symptoms in autism.

### 4.4. Limitations

This study has limitations. *First*, our estimation of GRSI, GR peak frequency, and developmental trends in GR parameters may be biased. Children who were less compliant and behaviorally more excitable did not provide good quality data and were excluded from the analysis. In addition, the accepted cut-off for GR reliability led to the exclusion from the analysis of children who had generally weak and unreliable GRs. The latter occurred more often with younger participants. For 'medium' and 'fast' velocity conditions, it was not always possible to estimate the peak GR frequency in children with strong velocity-related GR suppression. Therefore, our findings on GR frequency in these conditions mainly characterized children with relatively weaker velocity-related suppression of GR and, possibly, a higher E-I ratio in V1. *Second*, although our study strongly suggests that inhibition deficits in V1 do contribute to atypical visual motion processing in children with ASD, it does not allow us to draw conclusions about ASD-related differences in top-down control of visual motion sensitivity [94].

### 5. Conclusion

Although the role of the early visual cortex in visual motion perception atypicalities in ASD has recently been questioned by fMRI and modeling studies, our MEG findings strongly suggests that the poor directional sensitivity to small moving gratings observed in some autistic children is secondary to more basic, inhibition-related processes that influence directional

tuning of the receptive fields in the V1. Considering the large heterogeneity of neurophysiological mechanisms affecting the E-I balance in ASD, this deficit may be present in certain neurophysiological subtypes of ASD and characterize specific molecular pathways associated with autism.

## Supporting information

**S1 Fig. Individual GR spectra.**
(HTML)

**S2 Fig. Violin plots of the GR power values in TD and ASD children.**
(EPS)

**S1 Table. Spearman correlations of psychometric variables (MPI IQ, SRS) with MEG and psychophysical parameters in children with ASD.**
(DOCX)

**S1 Data.**
(XLSX)

## Acknowledgments

We would like to thank children and their families for participation in this study. The study was conducted at the unique research facility "Center for Neurocognitive Research (MEG-Center)" of MSUPE.

## Author Contributions

**Conceptualization:** Elena V. Orekhova, Tatiana A. Stroganova.

**Data curation:** Ilia A. Galuta, Andrey O. Prokofyev.

**Formal analysis:** Elena V. Orekhova, Viktoriya O. Manyukhina, Tatiana A. Stroganova.

**Funding acquisition:** Elena V. Orekhova.

**Investigation:** Ilia A. Galuta, Andrey O. Prokofyev, Dzerassa E. Goiaeva, Tatiana S. Obukhova, Kirill A. Fadeev.

**Methodology:** Elena V. Orekhova, Tatiana A. Stroganova.

**Project administration:** Ilia A. Galuta, Dzerassa E. Goiaeva.

**Software:** Elena V. Orekhova, Viktoriya O. Manyukhina.

**Supervision:** Elena V. Orekhova, Tatiana A. Stroganova.

**Visualization:** Viktoriya O. Manyukhina.

**Writing – original draft:** Elena V. Orekhova, Viktoriya O. Manyukhina.

**Writing – review & editing:** Elena V. Orekhova, Viktoriya O. Manyukhina, Justin F. Schneiderman, Tatiana A. Stroganova.

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
