## [Decision Letter · Decision Letter 0]

8 Dec 2022

PONE-D-22-28959Gamma oscillations point to the role of primary visual cortex in atypical motion processing in autismPLOS ONE

Dear Dr. Orekhova,

Thank you for submitting your manuscript to PLOS ONE. After careful consideration, we feel that it has merit but does not fully meet PLOS ONE’s publication criteria as it currently stands. Therefore, we invite you to submit a revised version of the manuscript that addresses the points raised during the review process. The reviewer were mostly positive. There are some minor changes, esp. in the figures, for instance Fig 1 and 2 may include some raw data or event related activity in time domain which will help to make sense of the observed spectral changes. Fig 1a, middle panel is not clear. Also important info on the figure conventions and panels are missing in the captions. For instance, Figure 6 includes many panels (columns) and can be improved by including titles to the panels, and referring back to them in the text. 

We look forward to receiving your revised manuscript.

Kind regards,

Mehdi Adibi, Ph.D., M.Sc., B.Sc.

Academic Editor

PLOS ONE

Journal Requirements:

"We would like to thank children and their families for participation in this study. 

This study was supported by Russian Science Foundation (project #22-25-00419)."

"This study was supported by Russian Science Foundation (project # 22-25-00419, to EVO)." 

Additional Editor Comments:

Dear Editors,

The reviewer comments were mostly positive, I have also reviewed the manuscript. There is only a few points by one of the reviewers, and I agree that requires some adjustment to in the figures to include experimental procedure and raw data and averaged waveform and event related activity prior to spectral analyses. This will help to visualise and characterise the sort of changes in the activity (time domain) that may contribute to spectral changes. this can be integrated in fig 1 and 2.

In figure 1a, panels can be named individually, the panel in the middle is confusing. Figure captions at times do not provide enough information about the plots. For instance, panels (columns) in Fig 6.

Reviewers' comments:

Reviewer's Responses to Questions

**Comments to the Author**

1. Is the manuscript technically sound, and do the data support the conclusions?

Reviewer #1: Yes

Reviewer #2: Yes

2. Has the statistical analysis been performed appropriately and rigorously? 

Reviewer #1: Yes

Reviewer #2: Yes

3. Have the authors made all data underlying the findings in their manuscript fully available?

Reviewer #1: Yes

Reviewer #2: Yes

4. Is the manuscript presented in an intelligible fashion and written in standard English?

Reviewer #1: Yes

Reviewer #2: Yes

5. Review Comments to the Author

Reviewer #1: The focus of the present study, titled “Gamma oscillations point to the role of primary visual cortex in atypical motion

processing in autism” from Orekhova et al., was the relationship between stimulation-related modulations of MEG gamma oscillations and perception of visual motion in children with ASD.

Considering how for some children with ASD, it is difficult to tolerate MEG and MRI data collection procedures, this work can give useful data.

After reading this work I have no specific requests for the authors and I consider the work done so far good for publication.

Reviewer #2: 1.To unravel the role of surround inhibition in V1 vs top-down modulation of V1 the authors investigated the role of putative neural inhibitory deficits in atypical motion perception in children with ASD by exploring velocity-related modulation of the visual GR, using two separate sets of high-contrast drifting gratings, presented in two separate experiments (psychophysical and MEG). Experiments suggests that the poor directional sensitivity to small moving gratings observed in some autistic children is secondary to more basic, inhibition-related processes that influence directional tuning of the receptive fields in the V1. Authors conducted the experiments in a rigorous way and explained every step in a clear way, indicating also the limitation of this study. In the material and methods authors indicated that only TD and ASD boys were employed, I just suggest to explain in a brief sentence, why girls are not included (male to female ratio in ASD). Even if well explained in the text, I suggest to add a more detailed description also in figure legend 1.

2. the statistical analysis has been performed appropriately and rigorously.

3. all data underlying the findings described in their manuscript are fully available.

4. the manuscript is presented in an intelligible fashion and it is written in standard English.

6. PLOS authors have the option to publish the peer review history of their article (what does this mean?). If published, this will include your full peer review and any attached files.

Reviewer #1: No

Reviewer #2: No

---

## [Author Response · Author response to Decision Letter 0]

10 Jan 2023

Responses to Editor’s comments

The reviewer were mostly positive. There are some minor changes, esp. in the figures, for instance Fig 1 and 2 may include some raw data or event related activity in time domain which will help to make sense of the observed spectral changes. Fig 1a, middle panel is not clear.

Also important info on the figure conventions and panels are missing in the captions. For instance, Figure 6 includes many panels (columns) and can be improved by including titles to the panels, and referring back to them in the text. 

We are grateful to the Editor and Reviewers for the positive evaluation of our work. 

As suggested, we revised figure 1 by including the raw data. We also made captions over all the panels, which, we hope, made the content easier to understand. Since Figure 2 presents information similar to that in Figure 1b (right-hand panels), we believe that the inclusion of raw data in this figure may be redundant. We marked all panels in figure 6 by letters of the alphabet and refer to them in the text.

Responses to Reviewer’s comments

Reviewer #1: The focus of the present study, titled “Gamma oscillations point to the role of primary visual cortex in atypical motion processing in autism” from Orekhova et al., was the relationship between stimulation-related modulations of MEG gamma oscillations and perception of visual motion in children with ASD.

Considering how for some children with ASD, it is difficult to tolerate MEG and MRI data collection procedures, this work can give useful data.

After reading this work I have no specific requests for the authors and I consider the work done so far good for publication. 

We are grateful to the reviewer for the positive evaluation of our work. 

Reviewer #2: 

We are grateful to the reviewer for taking the time to review our manuscript and provide comments.

1. In the material and methods authors indicated that only TD and ASD boys were employed, I just suggest to explain in a brief sentence, why girls are not included (male to female ratio in ASD). 

We added the following text to the Methods section (lines 141-145):

We limited our sample to males because the ratio of males to females among people with ASD is very high (~4/1) [51], and the relatively small sample size of our study would not have allowed us to analyze female participants with ASD as a separate group. On the other hand, the putative differences in neural excitability between males and females with ASD [52] preclude combining them into a single sample.

2. Even if well explained in the text, I suggest to add a more detailed description also in figure legend 1.

Thank you for this suggestion. We added more information to the legend.

Fig 1. Overview of the data collection and analysis pipelines. (a) Psychophysical experiment. Large or small gratings moving left or right were presented for a short period of time. The subject had to indicate the direction of motion by pressing the corresponding button. In case of a wrong response the exposure time increased, while in case of two correct responses in a row it decreased (‘one-up—two-down staircase’). The minimum time required for the subject to detect direction of motion (i.e. duration threshold) of small and large gratings was defined as the average exposure time at all but the first two ‘reversals’ of the corresponding staircase. (b) MEG experiment. The subject was presented with circular gratings drifting toward the center at different speeds. Brain activity corresponding to baseline and visual stimulation intervals was localized using LCMV beamforming. Stimulation-related changes in gamma power (gamma response) were calculated as (stimulation-baseline)/baseline. The peak frequency and power of gamma response (GR) were analyzed at ‘maximal sources’ in the visual cortex. For a detailed description of the experimental procedures and analysis, see the Methods section.

---

## [Editor Report · Decision Letter 1]

26 Jan 2023

Gamma oscillations point to the role of primary visual cortex in atypical motion processing in autism

PONE-D-22-28959R1

Dear Dr. Orekhova,

We’re pleased to inform you that your manuscript has been judged scientifically suitable for publication and will be formally accepted for publication once it meets all outstanding technical requirements.

Kind regards,

Mehdi Adibi, Ph.D., M.Sc., B.Sc.

Academic Editor

PLOS ONE
---

## [Editor Report · Acceptance letter]

3 Feb 2023

PONE-D-22-28959R1 

Gamma oscillations point to the role of primary visual cortex in atypical motion processing in autism 

Dear Dr. Orekhova:

I'm pleased to inform you that your manuscript has been deemed suitable for publication in PLOS ONE. Congratulations! Your manuscript is now with our production department. 

Kind regards, 

on behalf of

Dr. Mehdi Adibi 

Academic Editor

PLOS ONE